# Real-world Video Adaptation with Reinforcement Learning

**Hongzi Mao** [1 2]   **Shannon Chen** [2]   **Drew Dimmery** [2]   **Shaun Singh** [2]   **Drew Blaisdell** [2]   **Yuandong Tian** [2]
**Mohammad Alizadeh** [1]   **Eytan Bakshy** [2]

## Abstract

Client-side video players employ adaptive bitrate (ABR) algorithms to optimize user quality of experience (QoE).We evaluate recently proposed RL-based ABR methods in Facebook's web-based video streaming platform. Real-world ABR contains several challenges that requires customized designs beyond off-the-shelf RL algorithms —we implement a scalable neural network architecture that supports videos with arbitrary bitrate encodings; we design a training method to cope with the variance resulting from the stochasticity in network conditions; and we leverage constrained Bayesian optimization for reward shaping in order to optimize the conflicting QoE objectives. In a week-long worldwide deployment with more than 30 million video streaming sessions, our RL approach outperforms the prior ABR algorithm.

## 1. Introduction

The volume of video streaming traffic has been rapidly growing in the recent years (Cisco, 2016; Sandvine, 2015), reaching almost $60\%$ of all the Internet traffic (Sandvine, 2018). Meanwhile, there has been a steady rise in user demands on video quality — viewers quickly leave the video sessions with insufficient quality (Dobrian et al., 2011). As a result, content providers are striving to improve the video quality they deliver to the users (Krishnan & Sitaraman, 2012).

Adaptive bitrate (ABR) algorithms are a primary tool that content providers use to optimize video quality subject to bandwidth constraints. These algorithms run on client-side video players and dynamically choose a bitrate for each video chunk (e.g., 2-second block), based on network and

video observations such as network throughput measurements and playback buffer occupancy. Their goal is to optimize the video's quality of experience (QoE) by adapting the video bitrate to the underlying network conditions. However, designing a strong ABR algorithm with hand-tuned heuristics is difficult, mainly due to hard-to-model network variations and hard-to-balance conflicting video QoE objectives (e.g., maximizing bitrate vs. minimizing stalls) (Yin et al., 2015).

Facing these difficulties, recent studies have considered using reinforcement learning (RL) as a data-driven approach to automatically optimize the ABR algorithms (Mao et al., 2017). RL optimizes its control policy based on the actual performance of past choices, and it is able to discover policies that outperform algorithms that rely on fixed heuristics or use inaccurate system models. For example, as explained in Mao et al. (2017), RL methods can learn how much playback buffer is necessary to mitigate the risk of stall in a specific network, based on the network's inherent throughput variability. In controlled experiments with a fixed set of videos and network traces, a number of prior work has shown promising results for RL methods (van der Hooft et al.; Claeys et al., 2013). However, it remains unknown how the RL-based methods compare to the already deployed heuristic-based ABR methods in large-scale, real-world settings, where generalization and robustness are crucial for good performance (Systems & Research, 2019).

In this paper, we present the deployment experience of *ABRL*, an RL-based ABR module in Facebook's production web-based video platform. In designing of ABRL, we found that off-the-shelf RL methods were not sufficient to address the challenges that we encountered when attempting to deploy RL-based control policies in real-world environments. To learn high-quality ABR algorithms that surpass the deployed heuristics, we had to design new components in ABRL's learning procedure to solve the following challenges.

First, videos in production have different available bitrate encodings, e.g., some videos only have HD/SD encodings, while other videos have a full spectrum of bitrate encodings. However, standard RL approaches use neural networks (Hagan et al., 1996) that provide fixed outputs both in the num-

---

[1]MIT Computer Science and Artificial Intelligence Laboratory
[2]Facebook. Correspondence to: Hongzi Mao <hongzi@mit.edu>, Eytan Bakshy <ebakshy@fb.com>.

*Reinforcement Learning for Real Life (RL4RealLife) Workshop in the $36^{th}$ International Conference on Machine Learning*, Long Beach, California, USA, 2019. Copyright 2019 by the author(s).

ber of bitrates and the corresponding bitrate levels (e.g., the third output always corresponds to 720P encoding). To represent arbitrary bitrate encodings, we design ABRL's neural network to output a *single* priority value for each bitrate encoding; and we repeatedly use the same copy of the neural network for all encodings of a video. This approach scales to any video ABRL serves and supports end-to-end RL training (§3.2).

Second, ABRL experiences a wide variety of network conditions and different video durations during training. This introduces undesirable variance since conventional RL training algorithms cannot tell whether the observed QoE feedback of two ABR decisions differs due to disparate network conditions, or due to the quality of the learned ABR policy. To cope with the stochasticity of network conditions, we isolate the rewards on the actual network trace experienced in a training session, using a recent technique for RL in environments with stochastic input processes (Mao et al., 2019). This approach separates the contributions of the ABR policy from the overall feedback, enabling ABRL to learn robust policies across different deployment conditions (§3.3).

Third, production ABR requires balancing and co-optimizing multiple objectives together (e.g., maximize bitrate and minimize stalls). But RL requires a single reward value as the training feedback. Prior work merges the multi-dimensional objectives with a weighted sum (Yin et al., 2015). In practice, since ABRL's goal is to outperform the existing ABR algorithm in every dimension of the objective, this does not amount to a specific, pre-defined tradeoff between different objectives. To determine the weights for different reward components, we formulate the multi-objective optimization problem as a constrained optimization problem (i.e., optimizing one objective subject to bounded degradation along other objectives). This allows us to use constrained Bayesian optimization (Letham et al., 2018) to efficiently search for reward weights which best meet top-line objectives (§3.4).

Lastly, for ease of understanding and ensuring safety, we translate ABRL's learned ABR policy into an interpretable form for deployment. Specifically, we realize from the policy visualization that the learned ABR algorithm exhibits approximately linear behavior in the observed state of network and buffer occupancy. Thus, we fit a linear function of network throughput and buffer occupancy to approximate ABRL's learned ABR policy (§3.5). Such translation degrades the average stall rate by $0.8\%$, but provides full interpretability for human engineers. This allows engineers to understand the policy well enough to verify the learn policy.

We run A/B tests that compare ABRL with the existing ABR algorithm on Facebook's web-based video streaming platform. In a week-long worldwide deployment with more than 30 million video streaming sessions (§4), ABRL outperforms the heuristic-based ABR policy by $1.6\%$ in average bitrates and reduces stalls by $0.4\%$. For video sessions with poor network connectivity, in which cases the ABR task is more challenging, ABRL provides $5.9\%$ higher bitrate and $2.4\%$ fewer stalls. For Facebook, even a small improvement in video QoE is substantial given the scale of its video platform, which consists of millions of hours of video watches per day (Wagner, 2016). In this scale, a fraction of a percent consistent reduction in video buffering is significant; each day, this would save years of video loading time in aggregate.

## 2. Background

We provide a review of the basic concepts of adaptive video streaming over HTTP. Videos are stored as multiple chunks, each of which represents a few seconds of video playback. Each chunk is encoded at several discrete bitrates, where a higher bitrate implies a higher resolution and thus a larger chunk size. The chunks are aligned for seamless transitions across bitrates, i.e., video players can switch bitrates at any chunk boundary without fetching redundant bits or skipping parts of the video.

When a client watches a video, the video provider initially sends the client a manifest file that directs the client to a specific source (e.g., a CDN) hosting the video and lists the available bitrates for the video. The client then requests video chunks one by one, using an *adaptive bitrate (ABR) algorithm*. These algorithms use a variety of different inputs (e.g., playback buffer occupancy, throughput measurements, etc.) to select the bitrate for future chunks. As chunks are downloaded, they are stored in the playback buffer on the client. Playback of a given chunk cannot begin until the entire chunk has been downloaded.

## 3. Design

In this section we describe the design of ABRL, a system that generates RL-based ABR policies to deploy in Facebook's production video platform. We start by describing the simulator that hosts RL training in the backend (§3.1). Next, we explain the RL training framework (§3.2), which includes the variance reduction (§3.3) and reward shaping (§3.4) techniques needed for this application. Finally, we describe how ABRL translates the learned ABR policy to deploy in the front end (§3.5). Figure 1 shows an overview.

### 3.1. Simulator

To train the ABR agent with RL, we first build a simulator that models the playback buffer dynamics during video streaming. The buffer dynamics are governed by the standard ABR procedure described in §2. Specifically, the sim-

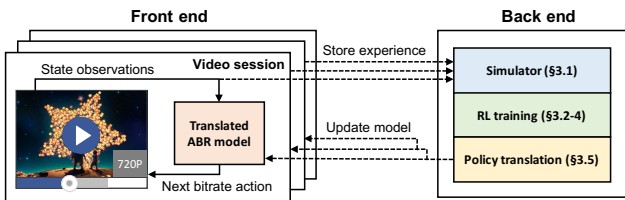

*Figure 1.* Design overview. For each video session in the production experiment, ABRL collects the experience of video watch time and the network bandwidth measurements and predictions. It then simulates the buffer dynamics of the video streaming using these experiences in the backend. After RL training, ABRL deploys the translated ABR model to the user front end.

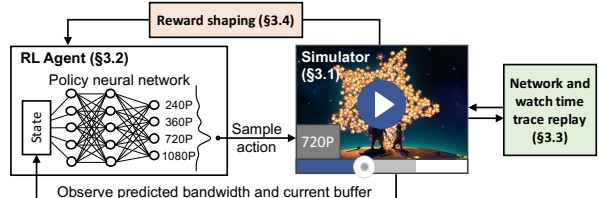

*Figure 2.* Backend RL training framework. ABRL updates the ABR policy neural network by observing the outcome when interacting with a simulator. The simulator uses production traces to simulate the video buffer dynamics.

ulator maintains an internal representation of the client's playback buffer, which includes the current size of buffer and the buffer capacity. The simulator invokes the ABR logic at each video chunk download event, where the ABR logic dictates the bitrate decision for the next chunk. For each chunk download, the simulator determines the download time based on the file size of the video chunk and the network throughput from the traces. Since the video is played in real time, the simulator then drains the playback buffer by the download time of the current chunk representing the video playback during the download. If the size of current playback buffer is smaller than the download time, we empty the buffer and issue a stall event. Subsequently, the buffer adds the duration of the downloaded chunk into the playback buffer. In the case where the buffer exceeds the capacity, the simulator ticks the time forward in the trace without downloading any chunk (i.e., move forward in the bandwidth trace). The simulated video session terminates at the end of each trace (corresponding to the end of a watch). During training, ABRL repeats the simulated video sessions by loading traces randomly at each time.

The simulator utilizes sampled traces collected from the actual video playback sessions from the frontend. At each video chunk download event, we log to the backend a tuple of (1) network bandwidth estimation, (2) bandwidth measurement for the previous chunk download, (3) the elapsed time of downloading the previous chunk and (4) the file sizes corresponding to different bitrate encodings of the video chunk. The bandwidth estimation is an output from a Facebook networking module. Note that the length of the trace varies naturally across different video sessions due to the difference in the watch time. In our training, we use more than 100,000 traces from production video streaming sessions.

### 3.2. Reinforcement Learning

The training setup shown in Figure 2 follows the standard RL framework. In this section, we describe the details of

the RL agent and the policy gradient training algorithm. In particular, we explain the challenges we encountered to motivate the variance reduction (§3.3) and the reward shaping (§3.4) techniques.

**RL setup.** Upon downloading each video chunk at each step $t$, the RL agent observes the *state* $s_t = (x_t, o_t, \vec{n}_t)$, where $x_t$ is the bandwidth prediction for the next chunk, $o_t$ is the current buffer occupancy and $\vec{n}_t$ is a vector of the file sizes for the next video chunk. As a feedback for the bitrate *action* $a_t$, the agent receives a *reward* $r_t$ constructed as a weighted combination of selected bitrate $b_t$ and stall time of the past chunk $d_t$:

$$r_t = w_b b_t^{v_b} - w_d d_t^{v_d} + w_c[\mathbb{1}(d_t > 0)], \quad (1)$$

where $\mathbb{1}(\cdot)$ is an indicator function counting for stall events, and $w_b, w_d, w_c, v_b, v_d$ are the tuning weights for the reward. Notice that these weights cannot be predetermined, because the goal of RL-based ABR is to outperform the existing ABR algorithm in every dimension of the metric (i.e., higher bitrate, less stall time and less stall count), which does not amount to a quantitative objective. In §3.4 we describe how we use Bayesian optimization to shape the weights for optimizing the multi-dimensional objective.

**Policy.** As shown in Figure 3, the agent samples the next bitrate action $a_t$ based on its parametrized policy: $\pi_\theta(a_t|s_t) \to [0, 1]$. In practice, since the number of bitrate encodings (thus the length of $\vec{n}_t$) varies across different videos (Lederer et al., 2012), we architect the policy network to take an arbitrary number of file sizes as input. Specifically, for each bitrate, the input to the policy network consists of the predicted bandwidth and buffer occupancy, concatenated with the corresponding file size. We then copy the *same* neural network for each of the bitrate encodings (e.g., the neural networks shown in Figure 3 share the same weights $\theta$). Each copy of the policy network outputs a "priority" value $q_t^i$ for selecting the corresponding bitrate $i$. Afterwards, we use a softmax (Bishop, 2006) operation to map these priority values into a probability distribution $p_t^i$ over each bitrate: $p^i = \exp(q_t^i) / \sum_{i=1}^{M} [\exp(q_t^i)]$. Importantly, the whole policy network architecture is end-to-end

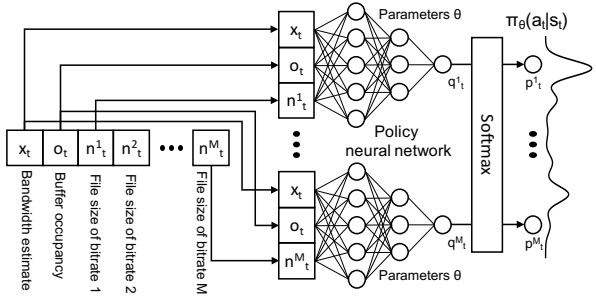

*Figure 3.* Policy network architecture. For each bitrate, the input is fed to a copy of the *same* policy neural network. We then apply a (parameter-free) softmax operator to compute the probability distribution of the next bitrate. This architecture can scale to arbitrary number of bitrate encodings.

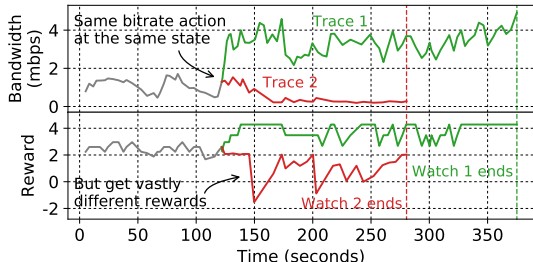

*Figure 4.* Illustrative example of how the difference in the traces of network bandwidth and video watch time creates significant variance for the reward feedback.

differentiable and can be trained with the policy gradient algorithms (Sutton et al., 1999).

**Training.** We use the policy gradient method (Sutton et al., 1999; Sutton & Barto, 2017; Tian et al., 2017) to update the policy neural network parameters in order to optimize for the objective. Consider a simulated video streaming session of length $T$, where the agent collects (state, action, reward) experiences, i.e., $(s_t, a_t, r_t)$ at each step $t$. The policy gradient method updates the policy parameter $\theta$ using the estimated gradient of the cumulative reward:

$$\theta \leftarrow \theta + \alpha \sum_{t=1}^{T} \nabla_\theta \log \pi_\theta(s_t, a_t) \left( \sum_{t'=t}^{T} r_{t'} - b_t \right), \quad (2)$$

where $\alpha$ is the learning rate and $b_k$ is a *baseline* for reducing the variance of the policy gradient (Weaver & Tao, 2001).

Notice that the estimation of the advantage over the average case relies on the accurate estimation of the average. For this problem, the standard baselines, such as the time-based baseline (Greensmith et al., 2004; Williams, 1992) or value function (Mnih et al., 2016), suffer from large variance due to the stochasticity in the traces (Mao et al., 2019). We further describe the details of this variance in §3.3 and our approach to reducing it.

### 3.3. Variance Reduction

ABRL's RL training on the simulator is powered by a large number of network traces collected from the front end video platform (§3.1). During training, ABRL must experience a wide variety of network conditions and video watches in order to generalize its ABR policy well. However, this creates a challenge for training: different traces contain very different network bandwidth and video duration, which significantly affects the total reward observed by the RL agent. Consider an illustrative example shown in Figure 4, where we use a fixed buffer-based ABR policy (Huang et al., 2014)

to make the bitrate action at time $\tau$. Even for this fixed policy, if the future trace happens to contain large bandwidth (e.g., Trace 1), the reward feedback will naturally be large, since the network can support high bitrate without stalls. In contrast, if the future network condition becomes poor (e.g., Trace 2), the reward will likely be lower than average. More importantly, the video duration determines the possible length of ABR interactions, which dictates the *total* reward the RL agent can receive for training (e.g., the longer watch time in Trace 1 leads to larger total reward). The key problem is that the difference across the traces is independent with the bitrate action at time $\tau$ — e.g., the future bandwidth might fluctuate due to the inherent stochasticity in the network; or a user might stop watching a video regardless of the quality. As a result, this creates large variance in the reward feedback used for estimating the policy gradient in Equation (2).

To solve this problem, we adopt a recently proposed technique for handling an exogenous, stochastic process in the environment when training RL agents (Mao et al., 2019). The key idea is to modify the baseline in Equation (2) to an "input-dependent" one that takes the input process (e.g., the trace in this problem) into account explicitly. In particular, for this problem, we implement the input-dependent baseline by loading the *same* trace (i.e., the same time-series for network bandwidth and the same video watch time) multiple times and computing the average total reward at each time step among these video sessions. Essentially, this uses the time-based baseline (Greensmith et al., 2004) for Equation (2) but computes the average return conditional on the specific instantiation of a trace. During training, we repeat this procedure for a large number of randomly-sampled network traces. As a result, this approach entirely removes the variance caused by the difference in future network condition or the video duration. Since the difference in the reward feedback is only due to the difference in the actions, this enables the RL agent to assess the quality of different actions much more accurately. In Figure 7, we show how this approach helps improve the training performance.

### 3.4. Reward Shaping with Bayesian Optimization

The goal of ABRL is to outperform the existing ABR policy according to multiple team-wide objectives (i.e., increasing the video quality while reducing the stall time). Recall that the reward weights in Equation (1) dictates the performance of ABRL's learned ABR policy in each of the objective dimensions. These objectives have an inherent trade-off: optimizing one dimension (by tuning up the corresponding reward weight) diminishes the performance in another dimension (e.g., high video quality increases the risk of stalls).

To determine the proper combination of the reward weights, we treat ABRL's RL training module (§3.2, §3.3) as a black box function $f(\vec{w}) \rightarrow (q, l)$ that maps the reward weights $\vec{w} \triangleq (w_b, w_d, w_c, v_b, v_d)$ from Equation (1) to a noisy estimate of the average video quality $q$ and stall rate $l$ in unseen evaluating video sessions, with known standard error $\sigma_q$ and $\sigma_l$, respectively.

Then, we use Bayesian optimization (Shahriari et al., 2016) to efficiently search for the weight combinations that leads to better $(q, l)$, with only a few invocations of the RL training module. This procedure of tuning the weights in the reward function is a realization of reward shaping (Ng et al., 1999). We formulate the multi-dimensional optimization problem as a constrained optimization problem:

$$\text{argmax}_{\vec{w}} \; q(\vec{w}), \; \text{subject to} \; \frac{l(\vec{w})}{l_s} \geq C \qquad (3)$$

Where $q(\vec{w})$ and $l(\vec{w})$ are the quality and stall rate evaluated at $\vec{w}$, $l_s$ is the stall rate of the status quo (non-RL based) policy, and $C$ is some constraint value.

Notice that the function $q(\cdot)$ and $l(\cdot)$ are can only be observed by running the RL training module—a computationally intensive procedure. We solve this constrained optimization problem with Bayesian optimization. Bayesian optimization uses a Gaussian process (GP) (Rasmussen, 2004) surrogate model to approximate the results of the RL training procedure using a limited number of training runs. Gaussian processes are flexible non-parametric Bayesian models representing a posterior distribution over possible smooth functions compatible with the data. We find that GPs are excellent models of the output of the RL training module, as small changes to the reward function will result in small changes in the overall outcomes. Furthermore, GPs are known to produce good estimates of uncertainty.

Using Bayesian optimization, we start from an initial set of $M$ design points $\{\vec{w_i}\}_{i=1}^{M}$, and iteratively test new points on the RL module according to an acquisition function that navigates the explore/exploit tradeoff based on a surrogate model (most commonly a GP).

A popular acquisition function for Bayesian optimization is expected improvement (EI) (see Frazier (2018, §4.1)). The basic version of EI simply computes the expected value of improvement at each point relative to the best observed point: $\alpha_{EI}(\vec{x} \,|\, f^*) = \mathbb{E}_{y \sim g(\vec{x} \,|\, \mathcal{D})}[\max(0, f(x) - f^*)]$, where $\mathcal{D} \triangleq \{\vec{w_i}, q(\vec{w_i})\}_{i=1}^{N}$ represents $N$ runs of data points, $f^*$ is the current best observed value and $g(\vec{x} \,|\, \mathcal{D})$ denotes the the posterior distribution of $f$ value from the surrogate.

We use a variant of EI—Noisy Expected Improvement (NEI)—which supports optimization of noisy, constrained function evaluations (Letham et al., 2018). While EI and its constrained variants (e.g., (Letham et al., 2018)), are designed to optimize deterministic functions (which have a known best feasible values), NEI integrates over the uncertainty in which observed points are best, and weights the value of each point by the probability of feasibility.

NEI naturally fits the structure of the optimization task, since the training procedure is stochastic (e.g., it depends on the random seed). We therefore evaluate the ABRL RL training module with a given $\vec{w}$ multiple times and compute its standard error, which are then passed into the NEI algorithm. NEI supports batch updating, allowing us to evaluate multiple reward parameterizations in parallel.

### 3.5. Policy Translation

In practice, most video players execute the ABR algorithms in the front end to avoid the extra latency connecting to the back end (Akhshabi et al., 2011; Sodagar, 2011; Adhikari et al., 2012; Huang et al., 2014). Therefore, we need to deploy the learned ABR policy to the users directly — i.e., the design of an ABR server in the back end hosting the requests from all users is not ideal (Mao et al., 2017). To massively deploy, we make use of the web-based video platform at Facebook, where the front end service (if uncached) fetches the most up-to-date video player (including the ABR policy) from the back end server at the beginning of a video streaming session.

For ease of understanding and maintenance in deployment, we translate the neural network ABR policy to an interpretable form. In particular, we found that the learned ABR policies approximately exhibit a linear structure — the bitrate decision boundaries are approximately linear and the distances between the boundaries are constant in part of the decision space. As a result, we approximate the learned ABR policy with a deterministic linear fitting function. Specifically, we first randomly pick $N$ tuples of bandwidth prediction $x$ and buffer occupancy $o$ (see the inputs in Figure 3). Then, for each tuple values $(x, o)$ and for each of the $M$ equally spaced bitrates with file sizes $n^1, n^2, \cdots, n^M$, we invoke the policy network to compute the probability of selecting the corresponding bitrate: $\pi(a^1 | x, o, n^1), \pi(a^2 | x, o, n^2), \cdots, \pi(a^M | x, o, n^M)$. Next,

we determine the "intended" bitrate using a weighted sum: $\bar{n} = \sum_{i=1}^{M} n^i \pi(a^i | x, o, n^i)$. This serves as the target bitrate for the output of the linear fitting function. Finally, we use three parameters $a, b,$ and $c$, to fit a linear model of bandwidth prediction and buffer occupancy, which minimizes the mean squared error over all $N$ points:

$$\sum_{i=1}^{N} \left| ax_i + bo_i + c - \bar{n}_i \right|^2. \tag{4}$$

Here, we use the standard least square estimator for the model fitting, which is the optimal unbiased linear estimator (Zyskind & Martin, 1969). At inference time, the front end video player uses the fitted linear model to determine the intended bitrate and then selects the maximum available bitrate that is below the intended bitrate.

Translating the neural network ABR policy provides interpretability for human engineers but it is also a compromise in terms of ABR performance (§4.2 empirically evaluates this trade-off). Also, adding more contextual based features would likely require a non-linear policy encoded directly in a neural network (§5). It is worth noting that directly using RL to train a linear policy is a natural choice. However, to our surprise, training ABRL with a linear policy function leads to worse ABR performance than the existing heuristics. We hypothesize this is because policy gradient with a weak function approximator such as a linear one has difficulty converging to the optimal, even though the optimal policy can be simple (Lu et al., 2018; Fujimoto et al., 2018; Fairbank & Alonso, 2012; Achiam et al., 2019).

## 4. Experiments

We evaluate ABRL with Facebook's web-based production video platform. Our experiments answer the following questions: (1) Does ABRL provide gains in performance over the existing heuristic-based production ABR algorithm? (2) How are different subgroups affected by the ABRL-based policy? (4) During training, how do different design components affect the learning procedure?

### 4.1. Overall live performance

In a week-long deployment on Facebook's production video platform, we compare the performance of ABRL's translated ABR policy (§3.5) with that of the existing heuristic-based ABR algorithm. The experiment includes over 30 million worldwide video playback sessions. Figure 5 shows the relative improvement of ABRL in terms of video quality and stall rate.

Overall, ABRL achieves a $1.6\%$ increase in average bitrate and a $0.4\%$ decrease in stall rate. Most notably, ABRL consistently selects higher bitrate through the whole week ($99\%$ confidence intervals all positive). However, choos-

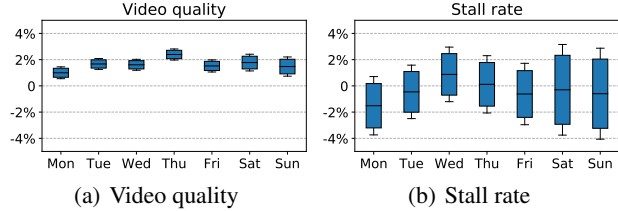

(a) Video quality        (b) Stall rate

*Figure 5.* A week-long performance comparison with production ABR policy. The comparison is sampled from over 30 million video streaming session. The box spans $95\%$ confidence intervals and the bars spans $99\%$ confidence intervals.

ing higher bitrates does *not* sacrifice stall rate — ABRL rivals or outperforms the default scheme on the average stall rate every day, even on Thursday when gains in video quality are highest. This shows ABRL uses the output from the bandwidth prediction module better than the fine-tuned heuristic. By directly interacting with the observed data, ABRL learns quantitatively how conservative or aggressive the ABR should be with different predicted bandwidths. As a result, this also leads to a $0.2\%$ improvement in the end-user video watch time.

These improvement numbers may look modest compared to the those reported by recent academic papers (Huang et al., 2014; Spiteri et al., 2016; Yin et al., 2015; Mao et al., 2017). This is mostly because we only experiment with web based videos, which primarily consist of well-connected desktop or laptop traffic, different from the prior schemes that mostly concern cellular and unstable networks. Nonetheless, any non-zero improvement is significant given the massive volume of Facebook videos. In the following, we profile the performance gain at a more granular level.

### 4.2. Detailed Analysis of RL Pipeline

**Reward shaping.** To optimize the multi-dimensional objective, we use a Bayesian Optimization approach for reward shaping (§3.4). The goal is to tune the weights in the reward function in order to train a policy that operates on the Pareto frontier of video quality and stall (and, ideally, outperform the existing policy in both dimensions). Figure 6 shows the performance from different reward weights during the reward shaping procedure. At each iteration, we set the reward weights using the output from the Bayesian optimization module, and treat ABRL's RL module as a black box, in which the policy is trained until convergence according to the chosen reward weights. The Bayesian optimization module then observes the testing outcomes (both video quality and stall) and sets the search criteria for the next iteration to be "expected improvement in video quality such that stall time degrades no less than $5\%$". As shown, within three iterations, ABRL is able to hone in on the empirical Pareto frontier. In this search space, there are many more weight

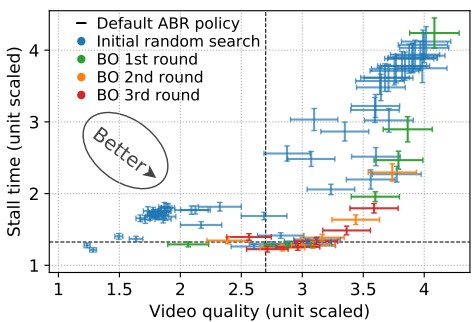

*Figure 6.* Reward shaping via Bayesian optimization using the ABRL simulator. The initial round has 64 random initial parameters. Successive batches of Bayesian optimization converge to optimal weightings that improve video quality while reducing stall rate. The performance is tested on held out network traces.

configurations that lead to better video quality (i.e., right of the dashed line) than the configurations leading to fewer stalls (i.e., lower than the dashed line). Compared to the existing ABR scheme, ABRL finds a few candidate reward weights that lead to better ABR policy both in terms of video quality and stalls (i.e., lower and to the right of the existing policy). For the production experiment in §4.1, we deploy the policies within the region that shows the largest improvement in stall. After this search procedure, engineers on the video team can pick policies based on different deployment objectives as well.

**Variance reduction.** To reduce the variance introduced by the network and the watch time across different the traces, we compute the baseline for policy gradient by averaging over the cumulative rewards from the same trace (in all the parallel rollouts) at each iteration, effectively achieving the input-dependent baseline (§3.3). For comparison, we also train an agent with the regular state-dependent baseline (i.e., output from a value function that only takes the state observation as input). Figure 7 evaluates the impact of variance reduction by comparing the learning curve trained with the input-dependent baseline to that with the state-dependent baseline. As shown, the agent with the input-dependent baseline achieves about 12% higher eventual total reward (i.e., the direct objective of RL training). Moreover, we find that the agent with input-dependent baseline converges faster in terms of the entropy of the policy, which is also indicated by the narrower shaded area in Figure 7. At each point in the learning curve, the standard deviation of rewards is around half as large under the input-dependent baseline. This is expected because of the large variance in the policy gradient estimation given the uncertainties in the trace. Fixing the trace at each training iteration removes the variance introduced by the external input process, making the training significantly more stable.

**Trade-off of performance for interpretability.** Figure 8

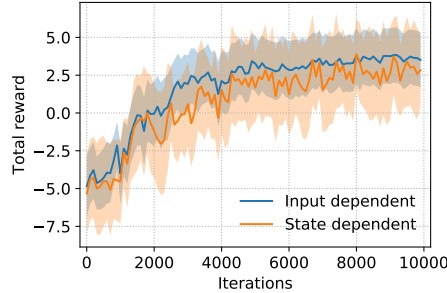

*Figure 7.* Improvements learning performance due to variance reduction. The network condition and watch time in different traces introduces variance in the policy gradient estimation. The input-dependent baseline helps reduce such variance and improve training performance. Shaded area spans ± std.

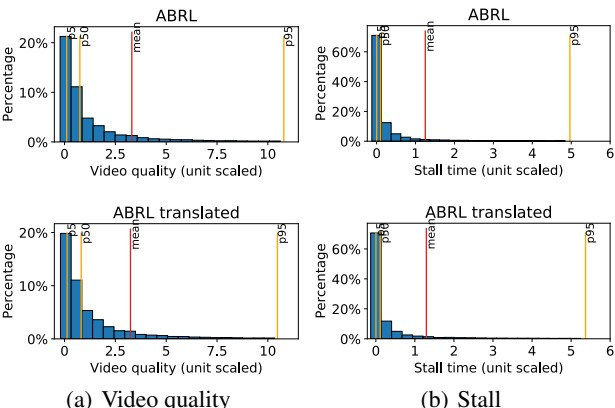

*Figure 8.* Performance comparison of ABRL and its linear approximated variant. The agents are tested with unseen traces in simulation. Translating the policy degrades the average performance by 0.8% in stall and 0.6% in quality.

shows how the testing performance of video quality and stall in simulation differ between ABRL's original neural network policy and the translated policy (§3.5). Most noticeably, making the ABR policy linear and interpretable incurs a 0.8% and 8.9% degradation in the mean and 95th percentile of stall rate. This accounts for the tradeoff to make the learned ABR policy fully interpretable. Also, we tried to train a linear policy directly from scratch (by removing hidden layers in the neural network and removing all the non-linear transformations). However, the performance of the directly learned linear policy does not outperform the existing baseline. This in part is because over-parametrization in the policy network helps learn a more robust policy (Lu et al., 2018; Fairbank & Alonso, 2012).

**Subgroup analysis.** To better understand how ABRL outperforms the existing ABR scheme, we breakdown the performance gain in different network conditions and we visualize the ABR policy learned by ABRL.

In Figure 9, we categorize the video sessions based on the

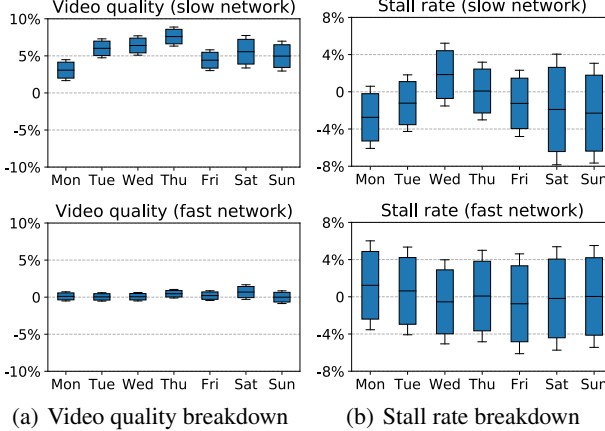

(a) Video quality breakdown      (b) Stall rate breakdown

*Figure 9.* Breakdown the performance comparison with different network quality for the live experiment. "slow network" corresponds to $< 500K$ mbps measured network bandwidth, and "fast network" corresponds to $> 10M$ mbps bandwidth. The box spans $95\%$ confidence intervals and the bars spans $99\%$ confidence intervals.

average measured network bandwidths. As shown, ABRL overall achieves a higher bitrate while maintaining fewer stalls in both fast and slow networks. Moreover, ABRL performs significantly better in slow network conditions, where it delivers $5.9\%$ higher bitrate with $2.4\%$ fewer stalls on average. When the network connectivity is unstable, ABR is challenging — a controller must agilely switch to lower bitrate when the bandwidth prediction or buffer level is low, but must avoid being too conservative by persistently sticking with low bitrates (when is is feasible to use higher bitrate without stalling). In the slow network condition, ABRL empirically uses the noisy network bandwith estimation better than the heuristic system in order to maintain better buffer levels. This indicates that ABRL optimizes algorithm performance under network conditions that existing schemes may overlook.

## 5. Discussion

We intend to work on several directions to further enhance ABRL in the production systems. First, ABRL's training is only performed once offline with pre-collected network traces. To better incorporate with the updates in the backend infrastructure, we can set up a continual retraining routine weekly or daily. Prior studies have shown the benefit of continual training with ever updating systems (Systems & Research, 2019).

Second, we primarily evaluate ABRL on Facebook's web-based video platform, because it has the fastest codebase update cycle (unlike mobile development, where the updates are batched in new version releases). However, the network

conditions for cellular networks have larger variability and are more unpredictable, where the gain of an RL-based ABR scheme can be larger (e.g., we observed larger performance gain for ABRL when the network condition is poor in §4.2. Developing a similar learning framework for mobile clients can potentially lead to larger ABR improvements.

Third, ABRL uses the same state variables (§3.2) as the current heuristic-based ABR algorithm. In practice, ABRL can also extend its state space to incorporate more contextual features such as video streaming regions, temporal information (which may contain different network characteristics) and continuous / many-valued features which engineers cannot easily fold into heuristics. Also, categorizing and optimizing the video quality based on video content types can likely result in better perceptual quality.

Lastly, there exists a discrepancy between simulated buffer dynamics and the real video streaming session in practice. Better bridging this gap can increase the generalizability of ABRL's learned policy. To this end, there is ongoing work addressing the discrepancy between simulation and reality with Bayesian optimization in reward shaping (Letham & Bakshy, 2019). Furthermore, another viable approach is to directly perform RL training on the production system. The challenge for this is to construct a similarly safe training mechanism (Alshiekh et al., 2018) that prevents the initial RL trials from decreasing perceptual quality of a video (e.g., restricting the initial RL policy from randomly select poor bitrates).

## 6. Conclusion

We presented ABRL, a system that uses RL to automatically learn high-quality ABR algorithms for Facebook's production web-based video platform. ABRL has several customized components for solving the challenges in production deployment, including a scalable architecture for videos with arbitrary bitrates, a variance reduction RL training method and a Bayesian optimization scheme for reward shaping. For deployment, we translate ABRL's policy to an interpretable form for better maintenance and safety. In a week-long worldwide deployment with more than 30 million video streaming sessions, our RL approach outperforms the existing carefully-tuned ABR algorithm by at least $1.6\%$ in video quality and $0.4\%$ in stall.

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
