# OpenReview forum: "Real-world Video Adaptation with Reinforcement Learning"
_ICML.cc/2019/Workshop/RL4RealLife — RL4RealLife 2019_

### Official Review · AnonReviewer3 · 2019-05-28
**Excellent work, interesting application**

**Rating:** 5
**Confidence:** 4

**Review:**

The paper is excellent in almost all aspects. This review won't belabour these, and will focus on potential issues, some of them may be future work.

Test data: It is great to test the algorithm in the FB production system for one week. It is desirable to evaluate its performance under conditions like holidays, special events like Super Bowl, even adversarial scenarios, etc.

State construction: The state construction follows a manual way, similar to manual feature engineering. Those chosen variables may be the most effective. However, as mentioned in Section 5 Discussion, it is desirable to evaluate state construction with more variables.

Exploration vs Exploitation: Any mechanism to study it in action selection? Is the softmax policy used in both training and testing/deployment? Or only the action with the highest probability is chosen in testing/deployment.

Off-policy learning: The proposed algorithm is trained with traces, thus facing issues with off-policy learning. It appears the paper does not discuss this.

Unbiasedness of baseline: It is desirable to discuss it (briefly). With a brief check of the reference Mao et al. 2019, it is discussed there.

Reward shaping: It is desirable to discuss if the reward shaping approach can guarantee the optimality of the policy without the reward shaping.

Policy translation: Why choosing these two variables? How about more variables?

Baseline was introduced in Williams' work (1987, 1992).

---

### Official Review · AnonReviewer2 · 2019-05-28
**solid paper**

**Rating:** 5
**Confidence:** 5

**Review:**

 This paper reports a real-world application of RL algorithms to video streaming.  The problem can be formalized as a sequential decision process where the decisions to make at each step concerns the quality of the video to deliver as well as the smoothness of the playing experience.

The authors identified three challenges in applying off-the-shelf RL algorithms to the domain:
 (1) various output dimensions, i.e., bitrate encodings could be sharply different with varying network conditions; (2) disparate input network conditions,  but feedback signal QoE did not reflect characterization of input; (3) dual and conflicting objectives:
 maximizing bitrates and minimizing stalls; (4) interpretability of results: treating NN model as black-box makes it difficult to debug, understand, and deployment to real-word scenarios.


To cope (1), the authors proposed a NN that outputs only a single value representing the
 "priority value" of the corresponding action choice; action selection policy is achieved by repeatedly feeding different actions into the same NN, and then apply a softmax function.

For (2), an "input dependent" baseline is used in the policy gradient objective.

For (3),  Bayesian optimization is adopted for a carefully-designed constrained objective.

For (4), the resultant NN is translated into linear model with certain sacrifice in accuracy
 but appropriation in interpretability.

With those techniques, the proposed method outperformed other RL algorithms in a large scale
 test on Facebook platform. The results look solid, described techniques are sensible. Overall this is a good application paper; I recommend to accept it.

---

### Decision · Program_Chairs · 2019-05-28

Accept